# Food hardness preference reveals multisensory contributions of fly larval gustatory organs in behaviour and physiology

Nikita Komarov[1], Cornelia Fritsch[1], G. Larisa Maier[1], Johannes Bues[2], Marjan Biočanin[2], Clarisse Brunet Avalos[1], Andrea Dodero[3], Jae Young Kwon[4], Bart Deplancke[2], Simon G. Sprecher [1]*

1 Department of Biology, University of Fribourg, Fribourg, Switzerland, 2 Laboratory of Systems Biology and Genetics, Institute of Bioengineering, School of Life Sciences, EPFL and Swiss Institute of Bioinformatics (SIB), Lausanne, Switzerland, 3 Soft Matter Physics Group, Adolphe Merkle Institute, University of Fribourg, Fribourg, Switzerland, 4 Department of Biological Sciences, Sungkyunkwan University, Suwon, Republic of Korea

* simon.sprecher@unifr.ch

**Data Availability Statement:** All data are fully available without restriction, All relevant data are within the manuscript and its Supporting

## Abstract

Food presents a multisensory experience, with visual, taste, and olfactory cues being important in allowing an animal to determine the safety and nutritional value of a given substance. Texture, however, remains a surprisingly unexplored aspect, despite providing key information about the state of the food through properties such as hardness, liquidity, and granularity. Food perception is achieved by specialised sensory neurons, which themselves are defined by the receptor genes they express. While it was assumed that sensory neurons respond to one or few closely related stimuli, more recent findings challenge this notion and support evidence that certain sensory neurons are more broadly tuned. In the *Drosophila* taste system, gustatory neurons respond to cues of opposing hedonic valence or to olfactory cues. Here, we identified that larvae ingest and navigate towards specific food substrate hardnesses and probed the role of gustatory organs in this behaviour. By developing a genetic tool targeting specifically gustatory organs, we show that these organs are major contributors for evaluation of food hardness and ingestion decision-making. We find that ablation of gustatory organs not only results in loss of chemosensation, but also navigation and ingestion preference to varied substrate hardnesses. Furthermore, we show that certain neurons in the primary taste organ exhibit varied and concurrent physiological responses to mechanical and multimodal stimulation. We show that individual neurons house independent mechanisms for multiple sensory modalities, challenging assumptions about capabilities of sensory neurons. We propose that further investigations, across the animal kingdom, may reveal higher sensory complexity than currently anticipated.

## Introduction

The properties of food play a crucial role in an animal's decision to ingest. While smell, taste, and visual properties provide important details about a food source, texture is a property of

Information files, as well as raw data available at Zenodo: 10.5281/zenodo.14216484 and NCBI Gene Expression Omnibus (accession number GSE149975).

**Funding:** This work was supported by the Swiss National Science Foundation grant 310030_219348 and IZKSZ3_218514 to SGS. The funder had no role in the study design, data collection and analysis, decision to publish, or preparation.

**Competing interests:** The authors have declared that no competing interests exist.

**Abbreviations:** AD, activation domain; BDSC, Bloomington Drosophila Stock Centre; DBD, DNA-binding domain; DO, dorsal organ; DOG, dorsal organ ganglion; GO, gustatory organ; GSN, gustatory sensory neuron; PCA, principal component analysis; PI, preference index; SEZ, subesophageal zone; TO, terminal organ; TOG, terminal organ ganglion; TRP, transient receptor potential; VDRC, Vienna Drosophila Resource Centre; VO, ventral organ; VOG, ventral organ ganglion.

food that is additionally critical. The texture of food serves as a multidimensional attribute of parameters not obviously determined by other sensory organs. Thus, the mechanical sensation of food sources is necessary for an animal's ability to completely evaluate the food it encounters.

The full extent of sensory roles of the *Drosophila* larval external sensory organs is not known. For example, although the presence of mechanosensory neurons in the primary taste sensing centre, the terminal organ (TO) was already suggested, identification of mechanisms, responses, and functions has proved elusive [1,2]. While it has been assumed that mechanosensation is important for decision-making, few studies have been conducted to elucidate the role of peripheral mechanosensation in the larva [1,3,4]; however, there is evidence that hardness of the food substrate affects food-related behaviour [5]. Meanwhile, the role of mechanosensation as a critical component of food decision-making in adults was recently characterised [6].

The perception of external cues is achieved by highly specialised sensory neurons. Different types of sensory neurons are thought to be tuned in a narrow fashion, thereby responding to a defined type of stimulation such as a specific range of wavelength of light or class of chemical compounds. Narrow tuning is assumed to be a critical feature of stimuli coding, allowing tightly regulated processing and integration in defined brain circuits. An essential function of taste systems revolves around distinguishing appetitive and aversive cues (e.g., "bitter" versus "sweet") at the level of the sensory neuron. Since this is the first point of contact with the chemical cue, a certain amount of debate is present about whether individual neurons can detect unique or multiple modalities. On the one hand, it is believed that neurons are either specifically or broadly tuned to one of 5 canonical tastes—sweet, bitter, umami, sour, and salt [7–9]. This is referred to as the "labelled-line" model. On the other hand, recent findings uncovered that individual taste neurons of both *Drosophila* larvae and mice are responsive to multiple modalities, including opposite hedonic valence [10–12]. In addition, recent findings in the mosquito and adult fly indicate that chemosensory neurons exhibit a varied receptor-type co-expression [13,14]. This indicates that the organisation, coding, and function of the peripheral chemosensory organs are more intricate than previously thought. Furthermore, the concept of an individual neuron, rather than the organ as a whole, integrating other senses such as light, mechanosensation, thermosensation, or hygrosensation has been suggested but remains to be explored [15].

The larva of the fruit fly *Drosophila melanogaster* provides a powerful model to uncover mechanisms of sensory perception due to its relative neuronal numerical simplicity, ample genetic tools, and, importantly, traceable processing and stereotypic behavioural responses [16]. Moreover, the larva represents a highly relevant model for exploring sensory systems and food consumption due to its biological need to ingest as much food of the highest quality possible. Failing to do so, the larva will either not undergo metamorphosis or develop into a smaller adult [17]. Larval taste is separated into external and internal components. On the exterior, the head of the larva bilaterally houses terminal organs (TOs)—the primary taste centre—and the dorsal organs (DOs)—the primary olfactory centre. Additionally, the ventral organ (VO) is also believed to be involved in taste, as well as other sensory modalities [18]. After ingesting food, larvae can taste food using pharyngeal sensilla, located along the oesophagus inside the mouth opening and projecting their dendrites into the gastrointestinal tract. Moreover, larvae are able to sense sugar not only in the sensory organs but also in the brain, where Gr43a, a receptor attributed to fructose sensing, is expressed, and this function is attributed to sensing the internal nutritional state of the animal [19].

The molecular basis of taste sensing is not fully understood. In the olfactory system, an individual *Odorant receptor* (*Or*) or *Ionotropic receptor* (*Ir*) gene is expressed alongside the obligate

*Odorant receptor co-receptor* (*Orco*) or one of 2 *Ir*-co receptors, respectively. In taste neurons, the organisation is different: *gustatory receptors* (*Gr*), *Irs*, and other putative chemosensors, such as the *pickpocket* (*ppk*) family, are co-expressed in an unclear manner [18]. Furthermore, the nature of Grs as channel-forming or signal-conducting proteins is not known, in contrast to the resolved tetramerisation of the OR complex in olfactory neurons. One exception is the $CO_2$-sensing complex comprised of Gr21a and Gr63a, which together confer carbonation sensing, but not either receptor alone [20]. Beyond this, a range of receptor genes have been proposed for specific modalities, such as *Gr66a* for bitter sensing or *ppk11* and *ppk19* for salinity [21–24]. Interestingly, despite sugar being a critical nutritional cue, no peripheral receptor has been identified. The canonical sugar sensor, *Gr43a*, is expressed in the pharyngeal sensilla and in the brain but not in the TO or VO [19]. Conversely, larvae are able to sense sugar at the periphery through multiple neurons extending their dendrites into the TO, albeit only one of these, C2, has a behavioural phenotype when silenced [11,12]. While being essential for larval survival and growth, the mechanisms for these responses have not yet been elucidated.

In order to study the role of the gustatory organs, we created a novel split-Gal4 line which drives reporter expression in the peripheral gustatory organs (GOs). By using behaviour assays, this tool allows us to demonstrate that these organs contribute not only to taste but also to mechanosensation. Additionally, by employing whole-organ and single-neuron volumetric in situ imaging, we show that individual neurons respond to chemical and mechanical stimuli. Furthermore, we show that one of these gustatory sensory neurons (GSNs) is multimodal, given its responses to both sugar and $CO_2$, as well as multisensory, and has the ability to respond to mechanical stimulation. Thus, we propose that multisensory integration in individual neurons may modulate their output, demonstrating a mechanism for context-based responses at the single-neuron level. Hereby, we show that a comparatively simple taste system integrates a significantly larger number of inputs than previously thought, which may account for a maggot's fascinating ability to distinguish a wide variety of taste stimuli.

## Results

### Larvae navigate to a specific range of substrate hardnesses corresponding to specific stages of fruit decomposition

While it has been reported that larvae prefer softer food substrates [4], we aimed to determine the range of preference exhibited by freely behaving animals. This would assess, for example, whether larvae will navigate to a harder unripe fruit (e.g., fresh apple) compared to a riper one (Fig 1A, top). Additionally, the compressibility of food could determine whether an animal will ingest it (Fig 1A, bottom). In order to evaluate traditional agarose-based experimental paradigms, we set out to understand how agarose concentration relates to physical properties of a flies' assumed natural food source—decaying fruit (Fig 1A'). Here, we tested apple, pear, banana, and pineapple fruits cut into similarly sized disks and allowed to decompose over 5 days (Fig 1A' and 1A"). We also note that the relative hardness of all fruits except apple is similar, suggesting a comparable relative rate of decay. Additionally, we observed that freshly cut fruits, except banana, are slightly albeit insignificantly harder than the highest (2.5%) agarose concentration in terms of the compression modulus (Fig 1B). Freshly cut banana fruit, however, strongly resembles the softness of 1% to 1.25% agarose concentration. It must be noted that this experiment presents a fundamental limitation in variability of fruit decay-mediated composition based on factors such as variety, storage conditions, transportation, among others. Thus, these results should only be used indicatively and not representatively.

Next, we assessed whether larvae will ingest foods across a variety of hardnesses. Here, we found a large percentage of larvae which readily ingest 500 mM sucrose-doped blue-dyed

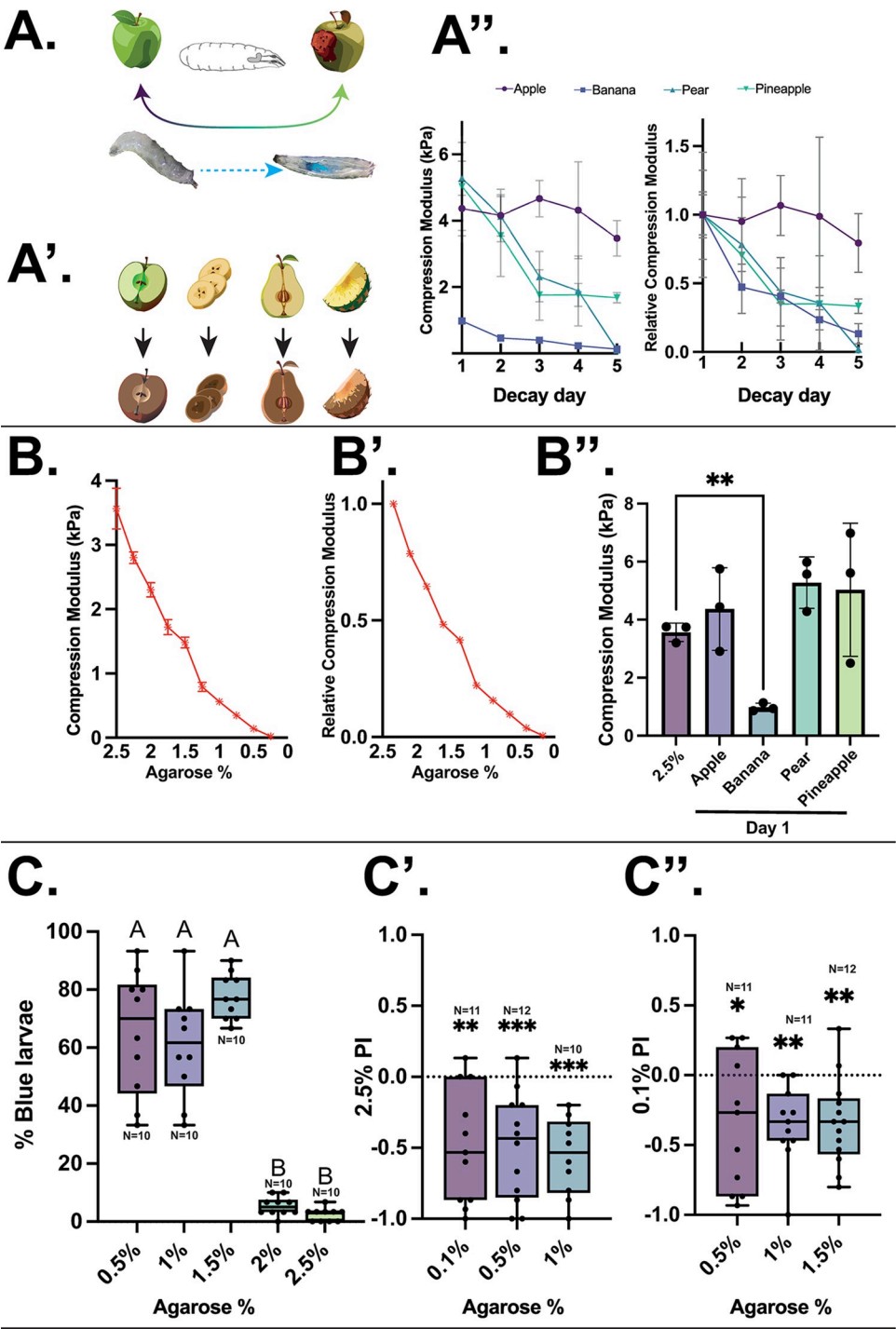

**Fig 1. Hardness preference in *Drosophila* larvae, as a relevant cue for varying feeding substrates.** (A) Top—cartoon of larval food environment, showing a harder (fresh) fruit and a softer (ripe) fruit. Bottom—larval ingestion can be visualised by blue-dyed agarose present and visible in the digestive system. A': Experimental paradigm involving a range of decaying fruit for mechanical analysis. A": Mechanical analyses of substrate properties of decomposing fruit (left) and hardness relative to the first measurement on day 1 identifying a decrease in hardness of dissected fruit over time, except for apple until day 5 after sectioning. (B) Compression modulus of a range of agarose concentrations showing a decrease in compression modulus in successively lower agarose concentrations. B': Relative hardness of agarose concentrations normalised to the mean 2.5% agarose hardness (error bars too small to visualise). B": Comparison of 2.5% agarose compared with freshly dissected fruit (day 1) showing similar compression modulus values for apple, pear, pineapple, and 2.5% agarose, with a significantly lower hardness value for freshly cut banana.

One-way Kruskal–Wallis with Dunn's multiple comparison tests, **-$p < 0.01$. (C) Grouped larval ingestion of a range of agarose concentrations after 2 min of exposure, presented as the percentage of larvae from a group of 30 per N containing blue-dyed agarose in the gut. Larvae readily and immediately ingest agarose food substrates up to and including 1.5%, however, cease to ingest beyond this threshold. One-way Kruskal–Wallis with Dunn's multiple comparison tests. Different letters (A or B) denote $p < 0.05$. C': Two-choice navigation preference between 2.5% agarose and 0.1%, 0.5%, and 1% agarose, respectively. Larvae show a consistent preference towards all softer agarose substrates compared to 2.5% agarose. One-sample $t$ and Wilcoxon test vs. 0. **-$p < 0.01$, ***-$p < 0.001$. C": Two-choice navigation preference between 0.1% agarose and 0.5%, 1%, and 1.5% agarose, respectively. Larvae show a consistent preference for the harder agarose concentrations. One-sample $t$ and Wilcoxon test vs. 0. *-$p < 0.05$, **-$p < 0.01$. $N = 10$–12 trials (×30 individuals) for all behaviour experiments. This figure is based on the data accessible at 10.5281/zenodo.14216484.

agarose substrate at concentrations of 0.5%, 1%, and 1.5% within the first 2 min (Fig 1C). The concentration of sucrose does not meaningfully affect this ingestion pattern when reduced (10 mM) or increased (1 M) (S4 Fig). However, at 2% agarose concentration and above, larvae almost entirely cease to ingest the substrate, indicated by the lack of identified larvae with blue dye present in the gut upon visual inspection. This indicates that there is a specific hardness threshold at which larvae are either unable or unwilling to ingest. Next, we determined whether larvae prefer softer or harder substrates, thus if hardness or softness presents as a specifically aversive or appetitive sensory cue for navigation by means of two-choice assays. Here, larvae were allowed to freely navigate on plates containing 2 areas of distinct agarose substrates. First, the larvae were given the choice between one half of the plate area containing 2.5% agarose and the other half containing one of 0.1%, 0.5%, or 1% agarose. We observed that larvae consistently prefer the softer concentration compared with 2.5% agarose (Fig 1C'). Next, larvae were given a choice between an excessively soft (0.1%) agarose substrate against 0.5%, 1%, and 1.5% concentrations. Here, we observe that larvae prefer to navigate onto slightly harder agarose (Fig 1C"). Thus, we observe that larvae appear to prefer the specific softness range between 0.5% and 1.5% agarose, with softer or harder food substrates being less preferential. Therefore, the hardness of the food substrate provides a specific sensory cue that allows the animals to navigate to optimal food substrates and preferentially ingest food of this hardness range. This correlates to the hardness of pear and pineapple after 3 to 4 days of decomposition, suggesting that hardness may present as an additional sensory cue.

## Understanding the role of gustatory organs in mechanosensation through the creation of a novel split-GAL4 driver

A basic way to investigate the function of a particular system is to inhibit or ablate it, subsequently observing the resulting phenotype. Thus, in order to investigate the GOs specifically, we required a driver that would allow for such manipulations. This would be similar to the role of *Orco*-GAL4 and Orco mutants in the dorsal organ ganglion (DOG), which were used to create anosmic animals effectively [25,26]. In order to decide on an approach for investing in the sensory system, we needed to understand the molecular profiles of the sensory neurons of the terminal organ ganglion (TOG) and the ventral organ ganglion (VOG) representing the GOs and the primarily olfactory DOG. Organs were dissected, digested, and subsequently sequenced using the deterministic, mRNA-capture bead and cell co-encapsulation dropleting system (DisCo, Bues and colleagues) [27] (S1 Fig). Through analysis of these data using the Seurat package [28,29], we isolated filtered objects expressing the neuronal markers *Neuroglian* (*Nrg*), *Synaptobrevin* (*Syb*), *Neuronal Synaptobrevin* (*nSyb*), and *pebbled* (*peb*), resulting in a set of 153 neurons. Moreover, in *pebbled*-positive cells, we identified that *Orco*, present in all olfactory cells of the DOG, does not overlap with cells expressing *proboscipedia* (*pb*), a member of the Hox transcription factor family known to mediate the specification of adult mouthparts

(Fig 2A, left) [30]. Using immunofluorescence staining, it emerged that Pb inclusively, but not exclusively, labels the neuronal population of the TOG (S1C Fig), while it is absent from the DOG. By more specifically targeting sensory neurons inserting the split-GAL4 components into the endogenous loci of the transcription factors Pebbled and Pb, we developed a specific split-GAL4 driver for the gustatory organs (Fig 2A, right; for details, see Materials and methods). Through immunohistochemical stainings and whole-mount in situ imaging, we determined that the driver covers the GOs but not the DOG (Fig 2B). Thus, we are able to drive reporters of our choosing in the peripheral taste organs specifically, giving us access to the well-established GAL4/UAS toolkit.

## Identifying the role of the GO contribution to sensory modalities

To uncover whether the broad range of cell types in the GOs results in a role in sensing different environmental stimuli, we selectively ablated neurons expressing both components of the *pb/peb* split-GAL4 by crossing these flies with the pro-apoptotic reporter UAS-*reaper* (*rpr*). Expectedly, we found that in a two-choice behaviour assay, the experimental larvae showed a significant reduction of response to both appetitive (sucrose) and aversive (quinine) agarose (Fig 2C). Additionally, we evaluated whether olfactory preference to an attractive odour (Ethyl acetate) or visual aversion to light would be affected and found no significant change (Fig 2C). Thus, we conclude that GOs are important for detecting gustatory cues, but are not critically required for driving light avoidance or olfactory attraction to ethyl acetate. To understand whether the range of proposed mechanosensory neurons of the GOs contributes to mechanical sensing, we used the paradigm of substrate hardness preference. Here, we observe that by ablating the GOs, the preference for "soft" substrate (1% agarose) is also significantly reduced, as well as the avoidance of a "very soft" substrate (0.1% agarose), which becomes slightly attractive (Fig 2C').

## Identification of mechanosensory gene expression in the larval head

Since mechanosensation has been proposed as a feature within the GOs previously, based on neuronal morphology [2], we set out to ascertain whether the proposed mechanosensory neurons in the primary sensory organs express canonical mechanoreceptor markers. We probed the scRNAseq data set and found that 3 genes involved in mechanosensory functions are present: *nanchung* (*nan*), *no mechanoreceptor potential C* (*NompC*), and *painless* (*pain*). The confidence for the expression was increased by means of immunofluorescence staining, finding 2 to 3 *nan*- and *NompC*-expressing cells in the larval head, along with a relatively broad expression of *pain* (Figs 3A and S2).

To more accurately pinpoint the principles of this mechanical sensing, we tested animals expressing RNAi for a range of known and putative mechanosensory genes, including those identified to be expressed in the peripheral chemosensory organs (Fig 3A). Here, we found that silencing expression of the TRPA family member *painless*, but none of the other candidates, results in a reduction of soft substrate preference akin to silencing the entire organ (Fig 3B). Furthermore, we tested whether hardness sensing contributes to ingestion decision-making by visually ascertaining the presence of blue-dyed agarose inside the animals. Previously, it was assumed that animals are unable, rather than unwilling, to ingest harder agarose as quickly as soft [11]. However, we found that ablating the GOs increases immediate ingestion (Fig 1) and similarly to softness preference, driving *painless* RNAi results in a similar phenotype, whereby the animals more readily ingest harder agarose (Fig 3B'). This suggests that *painless* (*pain*)-expressing neurons play a part in informing the animal about the hardness of the food.

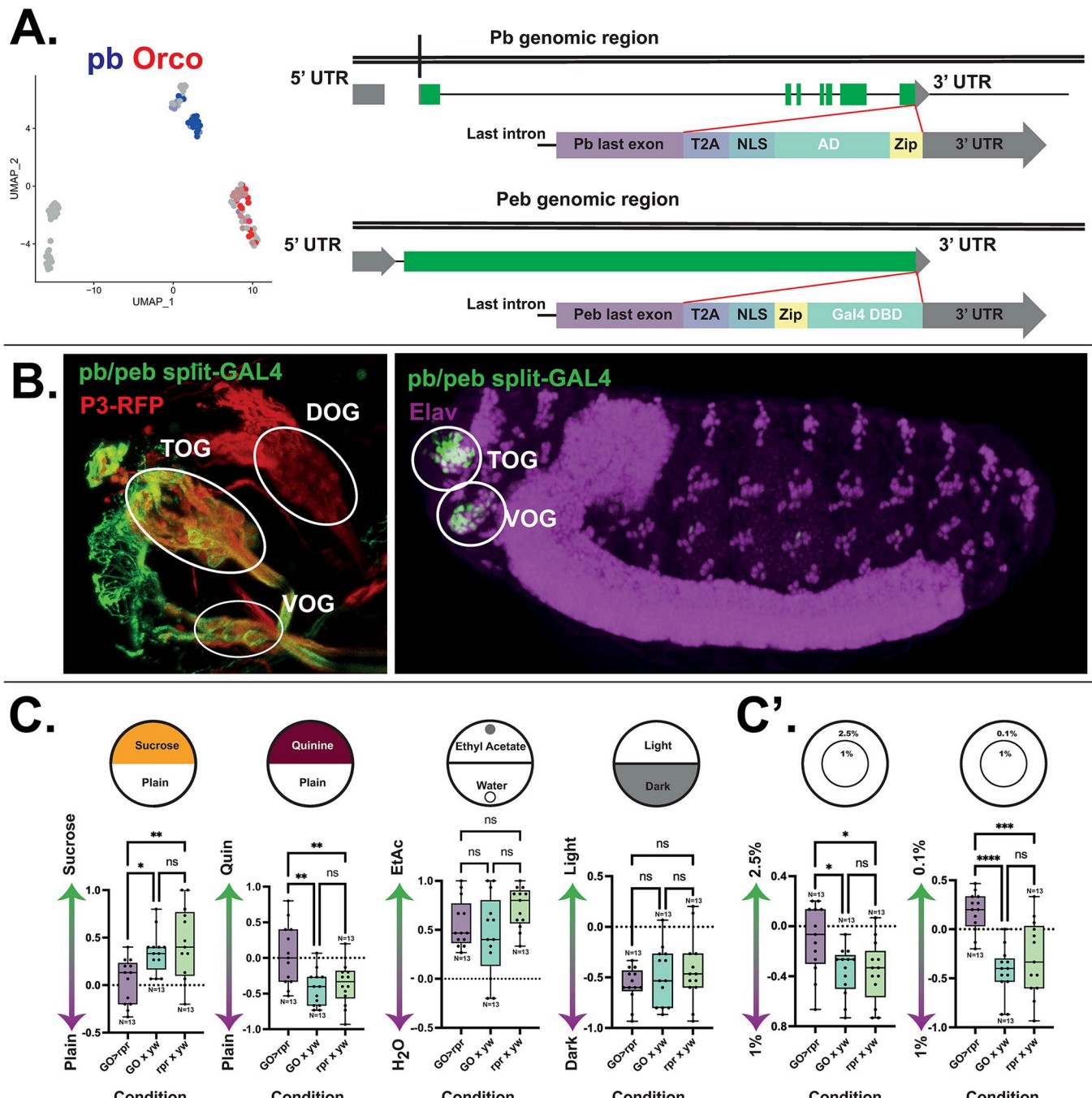

**Fig 2. Larval taste organ identification and characterisation of larval taste organ function.** (A) Left: neuronally filtered single-cell sequencing expression of the olfactory marker, *Orco*, and of *proboscipedia* (*pb*) within *pebbled*-positive cells. Right, cloning strategy for creation of a split-GAL4 knock-in. Upper: the activating domain (AD), along with a T2A, NLS and zipper (Zip) domains was inserted between the last exon of Proboscipedia and the 3′ UTR. Lower: the GAL4 DBD was inserted along with T2A, NLS, and Zip domains between the last exon of *pebbled* and the 3′ UTR. (B) Immunofluorescence stainings showing the expression of the transgenic 3xP3-RFP and generated *pb/peb*-split-GAL4 driving a UAS-*myrGFP* reporter in the taste organs, but not the DOG (left). With expression in the embryonic phase (right) mirroring that of the larva. (C) UAS-*reaper* (*rpr*)-mediated ablation results in defects of appetitive (sucrose) and aversive (quinine) choice. Appetitive olfactory response to Ethyl acetate was unaffected in the ablated condition, as was light-aversive behaviour. C': UAS-*rpr*-mediated ablation of GOs results in defective substrate hardness preference. Larvae lose preference for the softer 1% agarose versus harder 2.5% agarose. In addition, larvae with ablated gustatory organs begin to preferentiate excessively soft (0.1%) agarose substrate. $N$ = 10–15 for all behaviour assays. One-way ANOVA (Tukey's multiple comparisons) test. Black (above): *-$p < 0.05$, **-$p < 0.01$, ***-$p < 0.001$, ****-$p < 0.0001$, ns—not significant ($p > 0.05$). Not significant where not shown. This figure is based on the data accessible at 10.5281/zenodo.14216484 and NCBI GEO (accession number GSE149975). AD, activation domain; DBD, DNA-binding domain; DOG, dorsal organ ganglion; GO, gustatory organ.

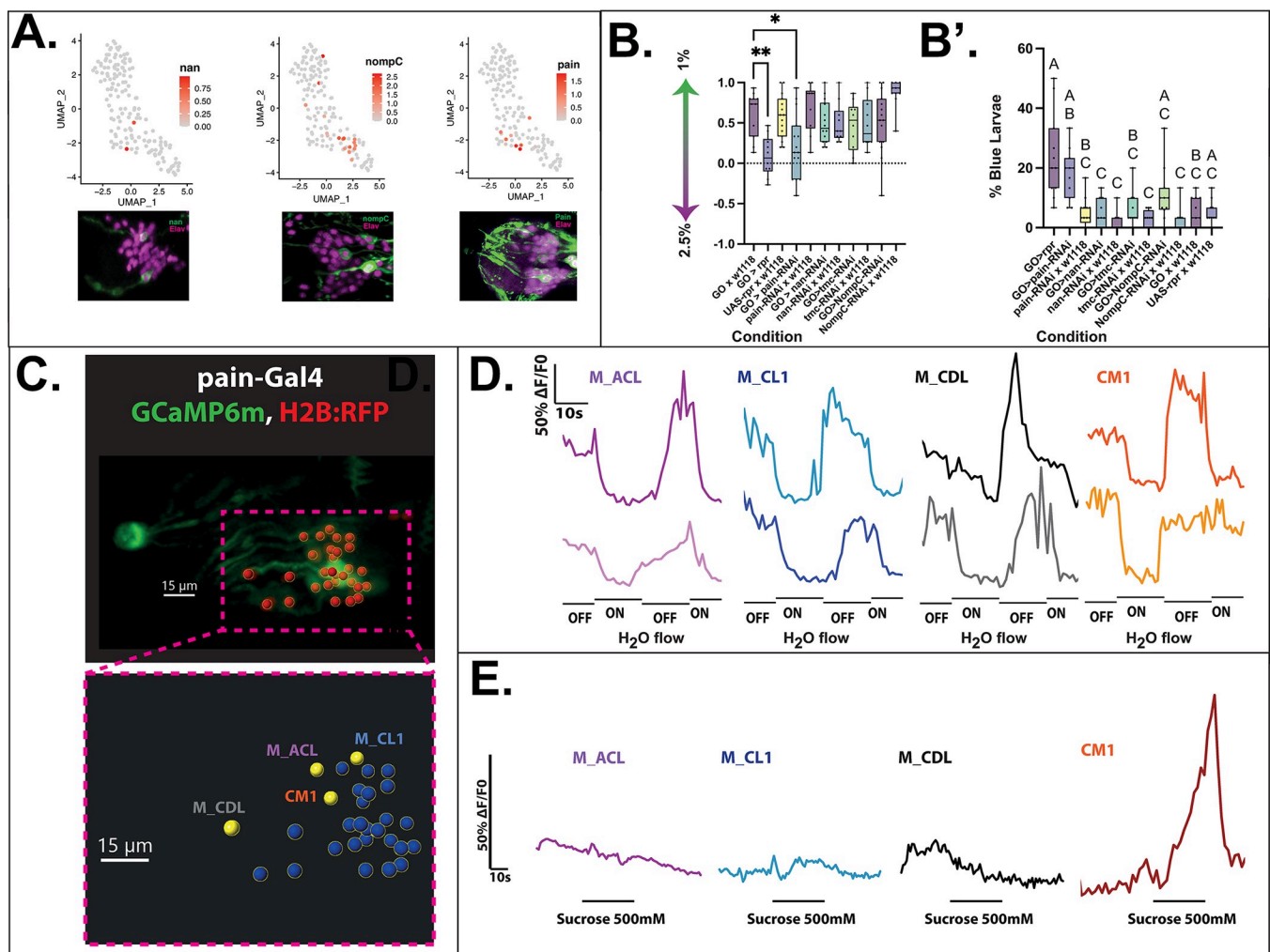

**Fig 3. Evaluation of genetic and physiological characteristics underlying mechanosensory function.** (A) Single-cell RNA sequencing and immunohistochemical stainings showing the expression of the mechanosensory genes *nanchug* (*nan*), *nompC*, and *painless* (*pain*). In immunohistochemical stainings, *nan* shows expression in 2 neurons of the head organs, *nompC* is expressed in 3 neurons, and a broad expression of *pain* can be observed. (B) Panel testing of putative mechanoreceptor genes shows that expressing *painless*, but not *nan*, *tmc*, or *nompC* RNAi results in a softness preference defect similar to complete Rpr-mediated ablation of GOs, with significance level approaching that of the common significance threshold (*p* = 0.0527). B': Ingestion of blue-dyed hard (2.5%) agarose by GO and $w^{1118}$ larvae crossed with a set of mechanosensory RNAi-knockdown reporters. While control larvae ($w^{1118}$ x UAS-rpr) do not ingest the hard agarose, larvae expressing *painless* RNAi in the GO show a greater degree of ingestion. *N* = 10–15 for all behaviour assays. One-way ANOVA (Tukey's multiple comparisons) test. Different letters denote *p* < 0.05. Not significant where not shown. (C–E) In situ calcium imaging view of 4 neurons in the GO expressing UAS-GCaMP6m (green) and UAS-H2B::RFP (red) under the control of pain-Gal4 (C, highlighted in yellow in inset) across 2 recordings which show relatively similar mechanosensory traces (D). One neuron, Centro-medial 1, shows responses to both mechanical stimulation and chemical stimulation (sucrose, 500 mM) (D, E). This figure is based on the data accessible at 10.5281/zenodo.14216484 and NCBI GEO (accession number GSE149975). GO, gustatory organ.

### Identification of TOG neurons physiologically responding to mechanical stimulation

Using volumetric calcium imaging recordings, we tested whether applying a shear force by switching on and off of water flow through a microfluidic chamber (i.e., applying pressure) elicits a response in *pain*-expressing neurons. Here, we noted that 3 to 4 neurons, varying by each experiment, in the TOG respond to this stimulus with a reduction of fluorescence, suggesting a "silencing" effect of mechanical stimulation (Fig 3C and 3D). To test whether any of the responding neurons also carry a chemosensory role, we applied a sugar solution due to its

broad and characterised response profile [11] to ascertain the presence or absence of multisensory responses. Intriguingly, we found that only 1 *pain*-expressing neuron, which we named central-medial 1 (CM1) due to its anatomical position, responds to both mechanical stimulation and sucrose dynamically opposingly (Fig 3E). These responses were verified to be due to change in the fluorescence of GCaMP, rather than due to changing morphology of the neuron, by means of measuring fluorescence from the nuclear RFP marker used to map GFP fluorescence (see Materials and methods). Here, we do not observe any significant change in fluorescence (S3A Fig).

Using this initial finding, and knowing the individual identity of the majority of the sucrose-sensitive neurons [2,31], we probed these neurons (C2, C5-7) using individual GAL-4 driver lines in an effort to identify CM1. While we did not observe mechanosensory responses in C2, C5, or C7, excitingly, we did observe a consistent response of C6, concordant with whole-organ imaging (Fig 4A and 4B).

C6 is characterised by the individual GAL-4 drivers of the *Gr21a* and *Gr63a* receptors. Notably, these genes are known to be conserved across Diptera as essential to sensing carbon dioxide ($CO_2$) [20,32–34]. However, there has been no characterisation of physiological responses to $CO_2$ at the single neuron level in the larva. Furthermore, C6 has previously been shown to be multimodal, responding to high salt concentration, sucrose, glycerol, and HCl among substances tested using a calcium imaging approach [12]. Using *Gr63a*-GAL4 as the driver for the expression of GCaMP6m, we performed recordings of the neurons before and during stimulation with aqueous $CO_2$. Expectedly, we found a solid and robust activation response to $CO_{2(aq)}$, but not to $HCO_3^-{}_{(aq)}$ control, thus ensuring that the response is due to molecular $CO_2$ and not to carbonate ions or due to morphology change (Figs 3C and S3B).

Further, we confirmed that the mechanism for carbonation sensing relies on *Gr21a* and *Gr63a* through selective knockdown of expression using RNAi. Interestingly, in these conditions, the responses to sucrose and mechanical stimulation remained present, although the assumed depolarisation following the cessation of the mechanical stimulation was attenuated. This suggests independent mechanisms for the varied responses (Fig 4D and 4D'), although the physiology of C6 may be affected by $CO_2$ receptor knockdown. We find that knockdown of *painless* does not significantly impact the mechanical response of C6 (Fig 4D'), suggesting that while *painless* is expressed, it is not the primary channel for physiological responses to mechanical stimulation. Thus, we show that C6 is a multimodal chemosensory neuron that responds to both carbon dioxide and sucrose while it also exhibits responses to mechanical stimulation. In this light, C6 appears to be the first identified neuron which combines responses to multiple chemical and sensory modalities. However, the receptor expression profile beyond Gr21a and Gr63a, such as the sucrose receptor, remains unknown. In addition, as there is a noted lack of mechanical or sucrose phenotype of C6 under $CO_2$-receptor knockdown, we propose that the mechanisms of physiological responses to tested stimuli are fully not linked to each other.

## Discussion

In this study, we determined a specific preference for food substrate hardnesses in the *Drosophila* larva model. We also identify that the larva primarily employs the gustatory organs as key mechanosensors for substrate hardness evaluation. We also show that the commonly used agarose concentrations for larval behaviour correspond to more advanced fruit decomposition stages, which get softer with time. Although notable, this specific finding must be regarded as anecdotal due to the inability to control the genetics and harvest timing of fruits used. Nevertheless, this provides an insight into the physical properties of decomposing fruits, and how these properties relate to agarose substrates commonly used in behavioural experiments.

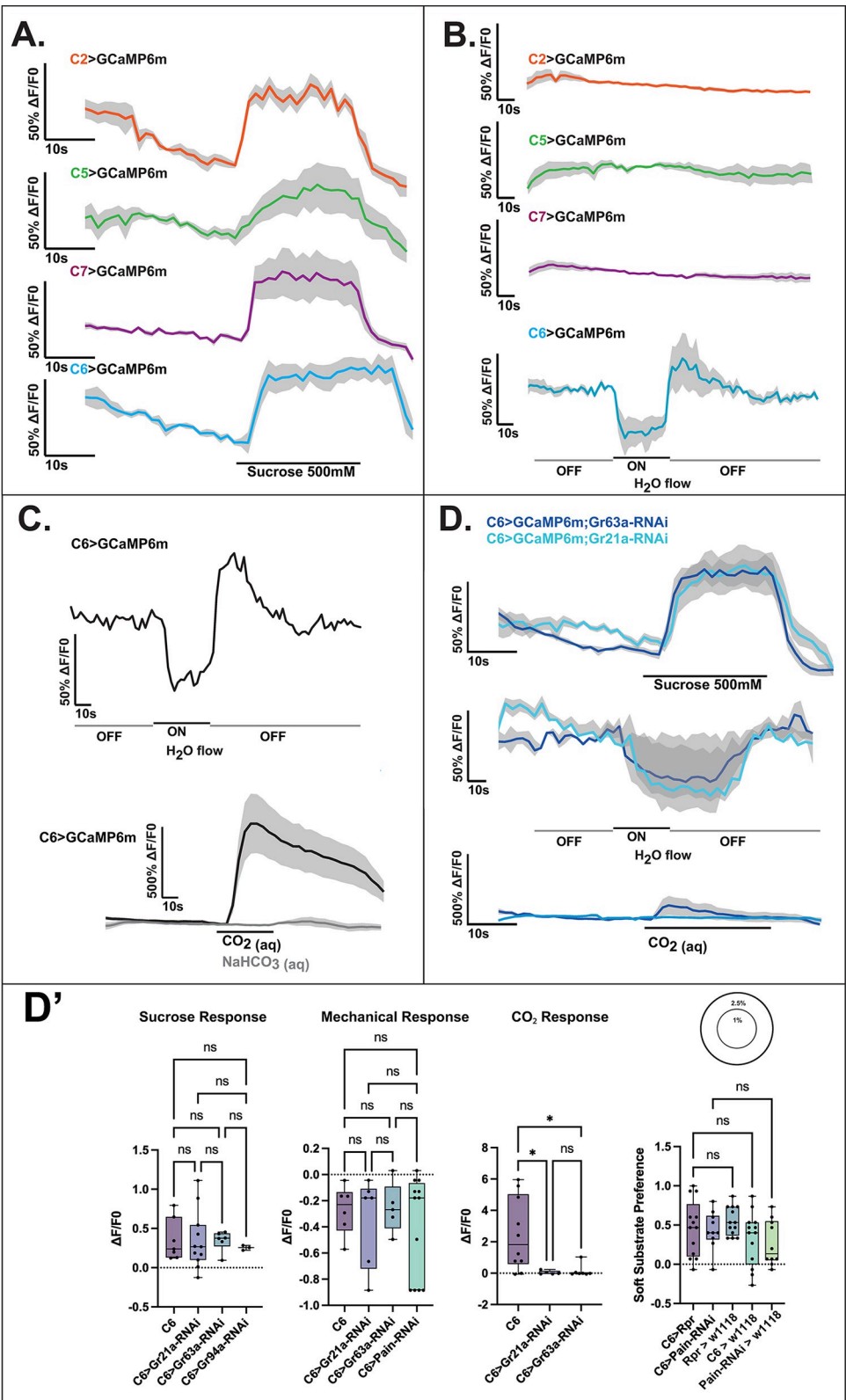

**Fig 4. Identification of a multisensory and multimodal neuron in the GO.** (A) Confirmation of single-neuron sucrose responses of comparable magnitude to CM1. C2, C5, C6, and C7 all show an approximately 50% fluorescence change when stimulated with sucrose. (B) Of the sucrose-sensitive neurons, only C6 shows a response to mechanical

stimulation. (C) Responses of C6 to mechanical stimulation and carbonated water, in an in situ calcium imaging paradigm with representative traces shown. C6 displays a strong and consistent response to $CO_2$, indicating that it bears a pseudo-olfactory role in carbon dioxide sensing. (D) D': RNAi-mediated gene knockdown of carbon dioxide receptors *Gr21a* and *Gr63a* in C6 shows that both are required for carbonation sensing, but not required for sucrose or mechanosensory sensing. However, the fluorescence signal profile appears to be affected by $CO_2$ receptor knockdown. Meanwhile knockdown of *painless* in C6 does not affect the calculated mechanical responses or soft substrate preference (D'). Shaded area denotes SD. $N = 10$ for all recordnigs. $N = 10$ for each bar, one-way ANOVA; *-$p < 0.05$, ns–not significant. This figure is based on the data accessible at 10.5281/zenodo.14216484. GO, gustatory organ.

Next, we present evidence of multiple sensory modalities being coded in model GOs. We show that GO ablation does not affect olfactory and light sensing but does affect taste and partial mechanical sensing. Additionally, we propose that the mechanoreceptor *painless*-expressing neurons affect the larva's ability to make food choice decisions, both for navigation (seeking) and ingestion. Thus, we show that the mechanosensory neurons contained within GO are sufficient in sensing the mechanical properties of the food substrate, and the repression of mechanosensory genes in these cells is sufficient for creating food-choice decision defects. We also show that independent mechanisms (i.e., not *Gr21a*/*Gr63a*) contribute to physiological responses to mechanical stimulation. Additionally, we show that while C6 is sensitive to mechanical force, it is not the sole contributor to mechanosensitive behaviours, as ablation of C6 does not affect behavioural preference to varying substrate hardness. This suggests that the signal transmitted by C6 is not relevant to this behaviour, or that other neurons are able to compensate in its absence.

While presumed before, the concept of mechanical perception being integrated within taste-sensing organs brings about fascinating questions about sensory integration in a numerically simple animal model. Finding that multimodal and multisensory neurons are present in the taste organs, we get further insight into the complexity of sensory processing [35]. While the mechanisms and functions behind multisensory responses remain mostly elusive, we identify that they are at least partially independent, mimicking similar findings in *C. elegans* and adult *Drosophila* [36–38]. We do, however, observe a longer period of inhibition of the C6 neuron upon knockdown of *Gr21a* or *Gr63a*, suggesting that the inherent physiology of the neuron may be affected upon the loss of these genes, specifically in how the neuron may return to a depolarised or resting state in contrast to the wild-type physiology. Further screening of gene expression in individual multimodal and multisensory neurons across models is required to understand their full mechanisms. However, we believe that this is the first demonstration of mechanical and chemical perception within an individual sensory neuron via independent mechanisms in *Drosophila*.

Interestingly, promiscuity of mechanoreceptors such as the transient receptor potential (TRP) family, where a wide variety of functions have been observed, from nociception to thermosensation across different models, may play a role in the varied mechanosensitive responses observed here [39–41]. Additionally, the co-expression of different mechanosensory genes, including the TRP family, within sensory neurons is also described [31,35,37], which, in coordination with our results, reveals an intriguing path for investigating individual receptor roles in the physiological responses of GSNs.

Moreover, these findings allow us to ask in-depth questions about the central processing of taste stimuli. It is possible to speculate that rather than transmitting information about an "appetitive" or "aversive" stimulus by individual neurons, as is the case for olfaction, the brain integrates the signals from the whole taste system before making decisions. That is, as multiple neurons sense the same stimuli, and yet the multimodal combinations are different, this creates a large sensory range when considering the number of unique combinations of neuronal

responses. Further, the recently released connectome data sets can allow for studies of local processing within the primary gustatory neuropil—subesophageal zone (SEZ), which, coupled with recent discoveries about peripheral sensory physiology as in this study, can shed more light on the logic of taste processing in the *Drosophila* larva and beyond. For example, one question that can be asked is whether different input modalities result in different signal outputs. For example, connectomic studies suggest that sensory neurons communicate with one another via axon-axonal interactions before they reach the brain or may result in outputs to synapses at different brain targets [42]. This could allow, for example, a neuron to modulate the signals perceived by the brain from its neighbours, which may, in turn, explain the reason for multimodality in individual gustatory sensory cells.

## Materials and methods

### Fly stocks and husbandry

Flies used for experimental crosses were maintained at 25°C on a 12/12 h dark-light cycle. Fly stocks were fed standard cornmeal food. Some commercially available lines were used in the study, including from the Bloomington Drosophila Stock Centre (BDSC) and the Vienna Drosophila Resource Centre (VDRC), please see S1 Table.

### Generation of a split-GAL4 line for gustatory organs

In order to generate a GO-specific split-GAL4 line, we chose 2 genes that are co-expressed in the cells of the taste organs but do not show overlapping expression in other tissues. We opted for the multiple zinc-finger transcription factor Pebbled (Peb) and the homeodomain transcription factor Proboscipedia (Pb) (Fig 1C). We decided to produce a split-GAL4 line expressing the DNA-binding domain (DBD) of GAL4 under the control of *peb* regulatory sequences and the p65 (GAL4) activation domain (AD) under the control of *pb* regulatory sequences. To ensure that the split-GAL4 constructs are expressed in the same cells as the endogenous genes, we fuse them in frame to the coding sequences of *peb* and *pb* using the CRISPR-Cas9 technique. To maintain the function of the endogenous transcription factors Peb and Pb and the split-GAL4 protein parts, the 2 proteins were connected with an autocatalytic peptide (T2A). After translation of the fusion proteins, the 18 amino acid-long T2A sequence will cleave itself just before its last amino acid, separating the endogenous protein from the attached split-GAL4 fragment and allowing both proteins to function independently. A protein zipper domain will combine both domains to a functional GAL4 complex in cells that express both fusion proteins. The *GAL4 DBD* fragment and zipper domain were PCR amplified from plasmid *pBPZpGAL4DBDUw* (Rubin Lab [43], addgene No. 26233) and cloned into *pBluescript*. The *T2A* sequence from plasmid *pF3BGX-T2A-p65-AD* [44] (Shu Kondo lab, addgene No. 138395) was added to the *zipper-GAL4DBD*. An 837 bp fragment of the C-terminal end of *peb* and 1,066 bp of its 3′ UTR were PCR amplified from genomic DNA isolated from *nos-Cas9* flies to serve as homology arms for the CRISPR template. Since the first CRISPR site used for insertion of the *GAL4* fragment is located within the *peb* coding sequence, the last 12 amino acids of Peb will be replaced with the T2A peptide after autocatalytic cleavage of the Peb-GAL4DBD fusion protein. The *p65AD* fragment, including *T2A* peptide, *NLS* and protein *zipper* domain, was PCR amplified from plasmid *pF3BGX-T2A-p65-AD* and cloned into *pBluescript*. A 1,434 bp fragment containing the last intron and last exon of pb and a 1,155 bp fragment containing its 3′ UTR and downstream genomic sequence, PCR amplified from *nos-Cas9* genomic DNA, were added as homology arms for the CRISPR template. In this construct, the entire Pb protein was fused to the T2A-NLS-GAL4 fragment, so that after autocatalytic cleavage the T2A peptide will be attached to the last amino acid of the Pb protein.

The *peb-GAL4DPD* template was injected into embryos of flies expressing Cas9 under the control of the *nanos* promoter (Bloomington stock no. 54591) along with a *pCFD4-U6:1_U6:3-tandemgRNAs* plasmid (Simon Bullock Lab [45], addgene no. 49411) expressing 2 gRNAs for CRISPR sites located at the end of the *peb* coding sequence and in its 3′ UTR. The *pb-p65AD* template was co-injected with a *pCFD4-U6:1_U6:3tandemgRNAs* plasmid expressing 2 gRNAs for CRISPR sites located in the 3′ UTR of *pb*. After eclosion, the *peb-GAL4DBD*-injected flies were crossed to a first chromosome balancer line (*N/FM7C-GFP*) and the *pb-p65AD*-injected flies were crossed to a third chromosome balancer line (*w;; Dr, e/TM3*). Single F1 offspring flies were crossed again with the appropriate balancer lines and PCR-screened for the presence of the *GAL4*-fragments. For each GAL4 fragment, 2 independent insertion lines were established. The 2 split-*GAL4* fragments inserted on chromosomes 1 and 3 were combined in a single line.

## Larval behaviour

Third instar, pre-wandering stage larvae were collected 72 to 96 h following crossing. Crosses were made on standard cornmeal food supplemented with liquid yeast paste. Animals were kept at 25°C on a 12 h/12 h dark-light cycle.

**Larval behaviour—Two choice assays.** Two-choice assays were performed as described thoroughly in Maier and colleagues [11], with the following modifications:

Hardness preference: 94 mm petri dishes were filled with either 0.1%, 1%, or 2.5% agarose boiled in nanopore water until dissolved. For 0.1% versus 1% assays, a 66 mm central circular cut-out was filled with the other concentration, respectively. A similar preparation was made for the 1% versus 2.5% assays. Plates containing the harder substance in the middle versus outside were randomly selected for each test. Thirty larvae were collected and rinsed in tap water before being placed on the centre of each plate. The number of larvae on each substrate was recorded at 2, 5, and 15 min. For larvae crossing the edge of the 2 substrate concentrations, the affirmative decision about the condition was made depending on the location of the mouth hooks. Following this, a preference index (PI) was calculated using the following formula:

$$PI = \frac{N \text{ larvae on softer substrate} - N \text{ larvae on harder substrate}}{N_{total} \text{ larvae}}$$

Light preference: 94 mm petri dishes were filled with 2.5% agarose. Thirty 3rd instar larvae per trial (crossed, maintained, and collected as above) were placed in the middle of the dish. Half of the dish lid was covered with aluminium foil, and the preparation was illuminated from a projector (Epson LCD Projector model H763B, default settings) positioned 70 cm above the experimental space, emitting a white (RGB: 255, 255, 255) light. The number of larvae on the illuminated side was counted, and a PI was calculated as follows to avoid exposure of the dark side to light:

$$PI = \frac{(N_{total} \text{ larvae} - N \text{ larvae on light side}) - N \text{ larvae on light side}}{N_{total} \text{ larvae}}$$

**Larval behaviour—Ingestion assay.** Petri dishes were filled with 2.5% agarose supplemented with Brilliant Blue dye (2% w/v). Thirty (30) 3rd instar larvae per trial (crossed, maintained, and collected as above) were placed on the blue agarose and left to wander for 2 min. Larvae were then collected, briefly washed in tap water. Larvae of different conditions were subsequently grouped, with each group labelled with a randomly assigned A, B, or C label. The randomly labelled groups were then handed over to the experimenter for quantification,

guaranteeing a double-blind experimental condition. The groups were examined for the presence of blue dye in the digestive tract, indicating ingestion. The percentage of larvae with blue dye present in the digestive tract was calculated using the following formula:

$$\frac{N\ blue\ larvae}{N\ total\ larvae} \times 100$$

**In situ calcium imaging.** In situ calcium imaging was performed as described in [46] and analysed as described in [11]. In short, L3 larval heads were dissected posteriorly to the brain and mounted inside a microfluidics chamber, and sealed in with 2% agarose in AHL saline. Water was pumped through the chip for the first 60 s of the recording, followed by a 30 s stimulation with tastant and a 30 s water wash. The following adjustments were made to the protocol:

Mechanical stimulation: To simulate mechanical pressure, larval heads were positioned within the microfluidic device and briefly washed with millipore water, with the flow being switched off 5 s before the start of the recording, with the larval head remaining in the aqueous environment of the microfluidic chamber. Macros were adjusted to switch the water flow on at the 60 s time point. Thus, corresponding neuronal responses were interpreted to result from shear stress (mechanical stimulation) rather than hygrosensation [47,48].

**In situ calcium imaging analysis.** Recordings from whole organ in situ calcium imaging, as well as experiments involving single neurons were analysed as described in detail in Maier and colleagues [11]. In brief, the recordings were imported into Imaris v9.6.0 where fluorescence values across time from tracked cell bodies was exported for downstream analysis (normalisation and background subtraction, plot generation) using R. For whole organ recordings, the recordings first underwent deconvolution using Huygens (Scientific Volume Imaging), followed by RFP signal mapping on the green channel. Fluorescence values were generated by tracing the GCaMP6m fluorescence values from the mapped nuclear (H2B:RFP) coordinates. Resulting fluorescence values across time were exported for downstream analysis in R, as above.

**Immunohistochemistry.** Embryos: Embryo collection, dechorionation, and fixation were performed as described in [49], with the 3.7% formaldehyde (Merck, 1.04003.1000) fixation method being employed before antibody incubation. Primary antibodies used: Rat-anti-Elav (DSHB 7E8A10, 1:50 dilution), Chicken-anti-GFP (Abcam 13970, 1:2,000 dilution), Mouse-anti-Pebbled (DSHB 1G9, 1:50 dilution), Rabbit-anti-DsRed (TaKaRa 632496, 1:1,000 dilution), Rabbit-anti-Pb (Cribbs, 1992 [50], 1:100 dilution). Fixed embryos were rehydrated in 1X-PBS (neoFroxx, 1346LT050) and briefly washed before incubation with primary antibodies in 1X-PBS with 0.3% Triton-X-100 (Roth, 3051.1) (PBS-T), overnight at 4˚C. The following day, the primary antibodies were removed, and the embryos were washed at RT with PBS-T for at least 2 h, replacing the PBS-T every 30 min. Secondary antibodies used: Donkey-anti-Mouse Alexa 488 (Molecular Probes A21202, 1:1,000 dilution), Donkey-anti-Rat Alexa 647 (Jackson ImmunoResearch 712-605-150, 1:1,000 dilution), Donkey-anti-Rabbit Alexa 647 (Molecular Probes A31573, 1:1,000 dilution), Goat-anti-Chicken Alexa 488 (Molecular Probes A11039, 1:1,000 dilution). Washed embryos were then incubated with the appropriate secondary antibodies in PBS-T overnight at 4˚C. The following day, the secondary antibody solution was removed, and embryos were briefly washed with PBS-T 3 times, before being washed in PBS-T with DAPI (Roth, 6335.2) (1:50,000 dilution) for 30 min at RT. The DAPI was then removed, and the samples were washed in PBS-T for at least 2 h, changing the PBS-T every 30 min. Following this, the PBS-T was replaced with mounting medium (90% Glycerol (Fischer

Scientific BP229-1), 0.5% N-propyl gallate (Sigma P3130), 20 mM Tris (Fischer Scientific BP152-5, pH 8.0) for at least 1 h at RT before mounting on standard glass slides.

L3 Larvae: Larval heads were dissected in PBS anterior to the mouth hooks, removing as much cuticle as possible without compromising the structural integrity of the samples. Dissected samples were kept in PBS on ice until the fixation step (dissection time should not exceed 1 h). Samples were then fixed in 3.7% formaldehyde in PBS for a minimum of 18 and a maximum of 30 min, shaking at RT. The formaldehyde solution was then removed, and the heads were briefly rinsed with PBS-T before being washed in PBS-T for at least 2 h, replacing the PBS-T every 30 min. Following this, the immunohistochemistry steps do not differ from those described for embryo labelling above. For improved penetration of the antibodies, 0.5% Triton X-100 was used.

Imaging and processing: Microscopy was carried out on the Leica Stellaris 8 Falcon confocal microscope, using the Plan APO 40×/1.10 water immersion objective. Acquired images were processed using Fiji ImageJ and figures arranged with Adobe Illustrator.

**Single-cell chemosensory cell suspension preparation for DisCo.**    Third instar larvae of genotype *nsyb*-Gal4 > UAS-*mcd8*::*GFP*; *Or83b*::*RFP* were collected from food, washed in tap water, PBS, dropped in ethanol and again PBS, and dissected in ice-cold PBS in such a manner that only the external chemosensory organs were kept, avoiding to include also the pharyngeal tissue containing internal chemosensory organs. The isolated material was placed on ice in elastase 1 mg/ml in siliconised 2-ml tubes. After dissecting 20 to 30 larvae (20 to 25 min), the tube sample was placed at room temperature to initiate digestion. After 30 min, the tissue was washed in PBS+BSA0,05% and dissociated by up-down pipetting 120 times using siliconised 200p pipette tips. Separated TOG and DOG (expressing *Or83b*::*RFP*), organs were detected using a fluorescence stereomicroscope and manually picked with a glass micropipette, placed in a final dissociation enzyme mix of Collagenase 1 mg/ml + Elastase 0.5 mg/ml for 10 to 15 min for single-cell suspension. The reaction was stopped with PBS + BSA 0,05%. Murine inhibitor was added at each step of the dissociation protocol.

**Deterministic co-encapsulation (DisCo) of chemosensory neurons for single-cell transcriptomics.**    Microfluidic chip design, fabrication and device handling are described elsewhere [27]. Following organ dissociation, target cell suspension was diluted in the cell loading buffer containing PBS 0.01% BSA (Sigma B8667), 6% Optiprep (Sigma D1556) and Murine RNase inhibitor (NEB M0314L) in the loading tip connected to the DisCo chip. After bead-cell in droplet co-encapsulation, sample droplets were transferred to a bead collection chip. Subsequently to bead capture, washing, reverse transcription (Thermo Scientific EP0753) and Exonuclease I (NEB M0293L) reactions were performed on-chip [51]. Beads containing cDNA were then eluted, and cDNA was amplified for 21 cycles using Kapa HiFi Hot start ready mix (Roche #07958935001). Amplified cDNA was then purified (GC biotech CPCR-0050) for quality assessment with Fragment Analyzer (Agilent). Libraries were then tagmented using in-house Tn5 [52], size selected and purified for sequencing on NextSeq 500 system (Illumina) following recommendations from original Drop-seq protocol [53] (20 bp for read 1 and 50 bp for read2) at sequencing depth above 400.000 reads per cell.

**Single-cell data pre-processing and analysis.**    The data analysis was performed using the Drop-seq tools package [53]. After pre-processing, reads were aligned to *Drosophila melanogaster* reference genome (Ensembl version 86) using STAR (version 2.7.0.e) [54]. Following the alignment, BAM files were processed using the initial package and read-count matrices were generated.

Downstream analysis was done using the Seurat package [29] version 3.1.2 in R version 4.2.2, in Rstudio version 2022.12.0+353. Individual data sets were loaded and used to create separate normalised and scaled Seurat objects of minimum 400 genes per cell. In order to

apply unique cell filters, we merged the data and then excluded cells with high gene numbers and high UMIs as potential doublets and cells with high mitochondrial gene percentages indicating potential apoptotic cells. Due to the observed correlation between cells with high gene number (nGene) and cells with high UMIs (nCount), by applying UMI threshold at 50,000 we also eliminated cells with more than 4,000 genes. Cells with mitochondrial gene percentages under 9% were kept for further analysis. Data was then integrated to circumvent batch effects using Seurat functions *FindIntegrationAnchors* and *IntegrateData*. As we were interested in characterising neurons, we used the *subset* function to keep only cells expressing *nSyb* or *peb* neuronal markers, excluding eventual surrounding tissue or cuticle cells. On the final data set of 153 neurons, principal component analysis (PCA) computation was followed by UMAP embedding, and clustering was performed at 0.5 resolution.

**Statistical analysis.**   Statistical testing and visualisation of data pertaining to in situ calcium imaging and single-cell sequencing was performed with R version 4.2.2 in R Studio version 2022.12.0+353. Quantitative analysis of behaviour and ΔF/F0 values was performed using Prism 9 (GraphPad Software), with bars showing min-max, with all points shown. The type of test, *p* values, and sample size for each graph are provided in the respective figure legends. Significance is displayed as follows: ns–not significant, $P < 0.05$(*), $P < 0.01$(**), $P < 0.001$(***), $P < 0.0001$(****). Figures were assembled using Adobe Illustrator 2024, including the use of the integrated generative AI feature for the creation of fruit icons in Fig 1.

**Substrate hardness evaluation.**   Compression tests were performed using an Anton Paar MCR 702 rheometer equipped with a CTD600 convection temperature device. A plate-plate geometry with an 8 mm diameter and a compression speed of 100 μm/s was used. All measurements were carried out at room temperature; 8 mm disks were cut from the samples and placed between the plates, applying a normal force of 0.1 N to ensure good sample loading.

Agarose plates were prepared in 50 ml of water at concentrations of 0.1% (gelling point), and in the range from 0.25% to 2.5% (w/v) in 0.25% increments by boiling the solutions for 2 min, with the volume adjusted to 50 ml post-boiling.

For fruits, 5 individual units of pear, apple, pineapple, and banana were cut into sections of ~5 mm thickness and then stored in a humidified incubator (100% RH, 25˚C) to simulate the decomposition process. For 5 consecutive days, 8 mm disks were cut from the sections and immediately measured.

The compression modulus was then calculated from the initial slope of the obtained stress-strain curves. Three measurements were repeated for each sample.

## Declaration of generative AI and AI-assisted technologies

During the preparation of this work, the authors employed the AI image generation feature in Adobe illustrator to create minor fruit graphics present in Fig 1. The authors have reviewed and edited the content as needed and take full responsibility for the published material.

## Supporting information

**S1 Fig. Single-cell sequencing procedure.** (A) Position of TOG and DOG within the larval head. (B) Isolation of TOG cells by fluorescent discrimination of GFP expressed in the pattern of the *Ir76b* promoter (subset of TOG neurons), and DOG through expression of Or83b::RFP (olfactory sensory neurons). B': Filtering of the cells by neuronal markers *Neuroglian* (*Nrg*), *Synaptobrevin* (*Syb*), *neuronal Synaptobrevin* (*nSyb*), and *pebbled* (*peb*). (C) Identification of olfactory neurons (marked by *Orco* expression) and taste neurons (marked by *Pb*) expression, showing no visible overlap of the 2 genes, as confirmed by immunohistochemical analysis. All cartoons and icons created manually using Adobe Illustrator. This figure is based on the data

accessible at 10.5281/zenodo.14216484 and NCBI GEO (accession number GSE149975).
(EPS)

**S2 Fig. Expression analysis of different receptor types in the larval head organs.** (A) Identified *Gr* genes. (B) Co-expression of specific *Gr* genes allows for identification of individual neurons, indicated by arrows. (C) Putative mechanoreceptor gene *pain* is co-expressed with *Gr66a* in at least 1 cell (arrow). (D) Identified *Or* genes. (E) Identified *Ir* genes. (F) Identified *Ppk* family genes. This figure is based on the data accessible at 10.5281/zenodo.14216484 and NCBI GEO (accession number GSE149975).
(EPS)

**S3 Fig. Fluorescence analysis of nuclear-expressed H2B:RFP in in situ imaging responses to stimulation.** (A) Traces showing neurons displayed in Fig 3D, displaying a lack of fluorescence change in the RFP channel to water flow activation and inactivation. (B) Stimulation of the C6 neuron as measured from the RFP channel, showing a consistent lack of significant change in fluorescence regardless of stimulus. $N = 5$–$9$, One sample $t$ and Wilcoxon test vs. 0 mean (left to right, $p = 0.79$, $p = 0.57$, $p = 0.47$). This figure is based on data that can be accessed at 10.5281/zenodo.14216484.
(EPS)

**S4 Fig. Sucrose-doped blue dyed agarose ingestion across a range of sucrose concentrations.** Ingestion of agarose measured by percentage of animals presenting with blue colorant in the gut after 2 min on substrate, groups of 30 larvae per N. (A) 10 mM Sucrose-doped agarose, (B) 500 mM Sucrose-doped agarose, (C) 1 M Sucrose-doped agarose. Regardless of sucrose concentration larvae readily ingest agarose food substrates up to and including 1.5%, however, ingestion is reduced at 2% and 2.5% agarose. One-way Kruskal–Wallis with Dunn's multiple comparison tests. Different letters denote $p < 0.05$. This figure is based on the data accessible at 10.5281/zenodo.14216484.
(EPS)

**S1 Table. Fly stocks used.**
(DOCX)

**S2 Table. Primers used for generation of the split-GAL4 lines (restriction sites underlined, CRISPR sites blue).**
(DOCX)

**S3 Table. One sample $t$ and Wilcoxon statistical analysis of GO ablation behavioural experiments compared with a 0 mean.**
(DOCX)

## Acknowledgments

We are also thankful to the University of Fribourg (UniFr) and the members of the Department of Biology for their support and help throughout our research. We especially thank Boris Egger and Felix Meyenhofer of the UniFr Bioimage Core Facility for their immense support in the imaging. We thank the Bloomington Drosophila Stock Centre and the Vienna Drosophila Resource Centre, Boris Egger and Shahnaz Lone for sourcing and sharing fly stocks.

We also thank all members of the Sprecher Lab for their invaluable feedback on our experimental design, procedures, and reporting. Their insights and suggestions are invaluable to our work.

Anton Paar GmbH is gratefully acknowledged for the loan of the MCR702 MultiDrive rheometer and their excellent technical support.

## Author Contributions

**Conceptualization:** Jae Young Kwon, Bart Deplancke, Simon G. Sprecher.

**Data curation:** Nikita Komarov.

**Formal analysis:** Nikita Komarov, G. Larisa Maier, Clarisse Brunet Avalos, Andrea Dodero.

**Funding acquisition:** Simon G. Sprecher.

**Investigation:** Nikita Komarov, Cornelia Fritsch, G. Larisa Maier, Johannes Bues, Marjan Biočanin, Clarisse Brunet Avalos, Andrea Dodero, Jae Young Kwon, Bart Deplancke, Simon G. Sprecher.

**Methodology:** Nikita Komarov, Cornelia Fritsch, G. Larisa Maier, Johannes Bues, Marjan Biočanin, Clarisse Brunet Avalos, Andrea Dodero, Bart Deplancke.

**Project administration:** Simon G. Sprecher.

**Resources:** Cornelia Fritsch, Jae Young Kwon, Bart Deplancke, Simon G. Sprecher.

**Supervision:** Simon G. Sprecher.

**Validation:** Nikita Komarov.

**Writing – original draft:** Nikita Komarov, Cornelia Fritsch, Simon G. Sprecher.

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
