## [Editor Report · Decision Letter 0]

3 Jul 2024

Dear Simon, 

Thank you for submitting your manuscript entitled "Food texture preference reveals multisensory contributions of gustatory organs in behaviour and physiology" for consideration as a Research Article by PLOS Biology.

Your manuscript has now been evaluated by the PLOS Biology editorial staff, as well as by an academic editor with relevant expertise, and I'm writing to let you know that we would like to send your submission out for external peer review.

Once your full submission is complete, your paper will undergo a series of checks in preparation for peer review. After your manuscript has passed the checks it will be sent out for review. To provide the metadata for your submission, please Login to Editorial Manager (https://www.editorialmanager.com/pbiology) within two working days, i.e. by Jul 05 2024 11:59PM.

Kind regards,

Roli

Roland Roberts, PhD

Senior Editor

PLOS Biology

rroberts@plos.org

---

## [Decision Letter · Decision Letter 1]

4 Sep 2024

Dear Dr Sprecher,

Thank you for your patience while your manuscript "Food texture preference reveals multisensory contributions of gustatory organs in behaviour and physiology" was peer-reviewed at PLOS Biology. It has now been evaluated by the PLOS Biology editors, an Academic Editor with relevant expertise, and by several independent reviewers. 

In light of the reviews, which you will find at the end of this email, we would like to invite you to revise the work to thoroughly address the reviewers' reports.

As you will see below, the reviewers are very interested in your study and overall supportive of publication. However, they also raise a few concerns that need to be addressed, including the need to be more precise with the physical properties you associate with texture.

Given the extent of revision needed, we cannot make a decision about publication until we have seen the revised manuscript and your response to the reviewers' comments. Your revised manuscript is likely to be sent for further evaluation by all or a subset of the reviewers.

**IMPORTANT - SUBMITTING YOUR REVISION**

*Re-submission Checklist*

*Published Peer Review*

*PLOS Data Policy*

*Blot and Gel Data Policy*

Sincerely,

Christian (on behalf of Roland, who is out of office this week)

Christian Schnell, PhD

Senior Editor

PLOS Biology

cschnell@plos.org

Roland Roberts, PhD

Senior Editor

PLOS Biology

rroberts@plos.org

REVIEWS:

Reviewer #1: Komarov et al. investigate the role of gustatory chemosensory neurons and mechanosensation in feeding in larval Drosophila. First, they describe that larvae prefer intermediate substrate hardness - they avoid very soft and very hard agar. To understand which taste neurons play a role in this behavior, they constructed a split GAL4-line, which specifically labels many gustatory receptor neurons in the fly larva. Ablating all these cells leads to deficits in sugar and quinine sensing but does not affect odor attraction and phototaxis. Larvae with ablated GRs also cannot differentiate substrate hardness anymore. Knocking down mechanosensors in the larvae revealed that painless receptors are required for hard substrate avoidance and proper feeding. Painless positive neurons also responded to mechanical stress and one of them also to sucrose. The authors identified the C6 neuron as being responsive to sucrose and mechanical stimulation. This neuron is also receptive to CO2 and expresses GR63a and GR21a. Ablating the CO2 GRs did not affect sucrose nor mechanical response, but the cells did not respond to CO2 anymore. Ablation of the C6 cell or knockdown of the painless receptor did not affect substrate preference, suggesting that other cells can compensate for the behavior.

The authors describe a new role for gustatory sensory cells, indicating that they are involved in substrate hardness sensing. They can further show that many GRs respond to mechanical stimulation and a broad ablation of GRs impairs softness sensing.

The current study provides further evidence that many chemosensory cells not only express a single receptor and might be broadly tuned to different cues. The authors should cite also the relevant literature for multisensory coding in a single sensory neuron in the adult fly:

- Vosshall lab, Herre et al., 2022 - mosquito sensory neurons express multiple sensors 

- Potter lab, Task et al., 2022 - ORNs express a multitude of receptors, such as IRs and GRs

The authors show that the C6 neuron responds to CO2, sucrose and mechanosensory stimulation, does it also respond to other tastants? Also, is it known which sensors mediate the sucrose responsiveness? 

Is the painless channel required for the neuronal response to mechanical stimulation in C6 - do they lose their response in functional imaging? This data would be relevant to show that painless is the softness receptor. However, other neurons might compensate for the behavioral stability even under C6 ablation (Figure 4D'). The authors do not describe this lack of phenotype in the results.

Reviewer #2: Komarov and colleagues perform genetic manipulations, calcium imaging and behavioral experiments to reveal the neurons and genes that allow Drosophila larvae to detect mechanical rigidity of their substrate, and that may be involved in integrating mechanical properties with other food related sensory cues. The conclusions about how mechanical perception may integrate with taste-sensing, including at the individual neuron level are interesting and seem well supported. I have major concerns with the framing of the work as investigating food "texture". 

MAJOR:

The use of the term "texture" in this work seems incorrect. More precise terminology is needed. For example, the different agar here has different material properties which are well described by terms such as compressibility, shear modulus, rigidity, elasticity or stiffness, but not texture as I typically understand it. I most associate the term texture with the surface profile of a material, so I would have expected the paper to explore the animal's ability to detect "bumpiness" or "roughness" of food surfaces: eg whether animals could detect small particles embedded in the agar that alter the agar's surface profile. Instead, the key experiments here all involve changes to compressibility or shear modulus of agar, while preserving its same surface profile. Indeed the authors use the term "hardness" and "softness" interchangeably with texture, which seems incorrect. I worry it is at best confusing and at worst outright incorrect to claim new findings about the neruobiological basis of detecting food "texture" based on the experiments presented here.

Similarly, the analogy to the human tongue probably has to be tempered. 

Minor:

- It seems overly strong to claim that "GOs do not appear to contribute to olfactory or light sensing" (line 223) after only testing out one odor (ethyl acetate). This claim should be tempered. 

-To help the reader it would be useful to add an x axis label in fig 1B and yaxis labels in figures 2C and 3B.

 -Also it would be helpful to specify that it is water that is on an off in the xaxis of figure 3EandD. 

Reviewer #3: Komarov and colleagues address a very interesting question regarding the contribution of food texture evaluation to feeding behaviour. This work clearly shows that the hardness of the agarose substrate affects food search and food intake behaviour. Further, the work identifies that mechanosensing neurons within the gustatory organ of the larva contribute to the food-choice decision. 

The manuscript makes an important contribution to our understanding of food preference and feeding behaviour and the underlying mechanisms, and is therefore undoubtedly worthy of publication in PLoS Biology. However, prior to publication, the authors should address a few issues that in some way weaken the current version of the manuscript.

Major comments:

1) I do not fully agree with the selection of genetic controls in a few of the individual experiments.

In Fig.2 two genetic controls are used: "GO control" and "rpr control" (I assume that the controls are GO/+ and rpr/+, the information is not provided) alongside with the experimental group "GO>rpr". That is a common standard in the field. However, Fig.3B provides data on different RNAi lines expressed in GO, but only w1118 is used as a control (not quite sufficient in my eyes). Further, in Fig.4D the authors decided to use GO (I assume GO/+, the information is not provided either) as a control in the first 3 panels, while in the last experiment of Fig.4D "GO control" and "rpr control" were used again. In my opinion the selection of genetic controls should not vary between the experiments and only the common standard (GAL4/+ and UAS/+) is sufficient for proper evaluation of the data. In this particular case, I don't think the results of the two data sets will change, but it would be beneficial to completely rule out an effect of the genetic background by the standard control.

2) The presentation of data in Fig.3C,D is not entirely convincing to me, although these results are of high importance. The authors state "we found that 3-4 neurons in the TOG respond to this stimulus with a reduction of fluorescence, indicating a silencing effect of mechanical stimulation". I have a few points here: Why 3-4? 4 neurons are presented in D-E. Please indicate why two curves are presented each (individual measurements?), but only one for CM1 - what is the difference here? Why is the time frame of the CM1 neuron so different (on-off duration) to the other neurons? Wouldn´t it be worthwhile to quantify the silencing effect (ΔF/F0) to see if there are differences in the effect depending on the type of neurons presented? L277: Why is only data shown on sucrose stimulation for CM1? It would be useful for the reader to see the data showing that M_ACL, M_CL1 and M-CDL do not respond to sugar. Why are single traces presented in Fig.3 (and Fig.4C), while data for the other imaging experiments are presented in mean + SEM(SD)? 

Minor comments:

1) Line 162-164: "Here, we observe that larvae appear to prefer the specific softness range between 0.5% and 1.5% agarose, with softer and harder substrates elicit aversion. Can the authors really argue that the conc. of agarose below 0.5% and above 1.5%, respectively, are indeed considered as "aversive stimuli"? This is an interesting observation; however, it might only be a preference for 0.5%-1.5% without the other concentrations be aversive. Could the authors provide any data on showing that aversion is triggered, like more head casts, higher crawling speed etc. on agarose conc. outside of the preferred range that would indeed point on aversion?

2) Figure 1C' elegantly shows that ablation of GOs reduces the preference for 1% over 2.5% agarose as well as the preference for 1% over very soft 0.1% agarose. However, it appears that larvae do not only lose the preference for 1%, but also show a slight preference for 0.1%, which would even illustrate a shift in valence. Could the authors provide statistical tests against chance level (whether larvae really prefer 0.1%) and comment on how this might fit into their model?

3) Figure 2C': Why was a different assay used here? The data presented in Fig.1 and Fig.2 C are based on a choice test where different concentrations of agarose are presented on a plate containing two halves of the substrates. However, Fig.2C' uses a different choice test with two circles inside each other. Are they comparable? Furthermore, the data are presented here as a preference for 1% over 2.5% (positive values), whereas they are presented as an avoidance of 2.5% in Fig.1B' (negative values), which is a little confusing.

4) L125-127: "We observed that freshly-cut fruits, except banana, are significantly harder than even the hardest (2.5%) agarose concentration in terms of the compression modulus". Figure 1A does not provide any indications for statistical comparisons. Please add.

5) L150-151: Please add more information about the amount/concentration of sugar added to the agarose.

6) L152 / Fig.1B: What does "% ingestion" mean? This is not clear to me.

7) L153-154: "This indicates that there is a specific hardness threshold at which larvae are either unable or unwilling to ingest it". Can the authors rule out that this is primarily true for the experiment done here with the specific concentration of sugar added to the substrate? What will happen if the conc. of sugar added is suboptimal? Does the threshold change?

8) Some references may be worth adding, e.g. Apostolopolou et al. 2014 (doi: 10.3389/fnbeh.2014.00011), as they show (in a slightly different context) that agarose concentration affects larval behavioural performance.

Reviewer #4: The authors of the paper "Food texture preference reveals multisensory contributions of gustatory organs in behaviour and physiology" investigate the role of neurons in the gustatory organ in integrating mechanosensory and taste information. First, they convincingly demonstrate a preference of larval Drosophila to navigate towards substrates with specific hardness, which matches their ability to ingest food with this hardness. They then show through cell ablation, RNAi and eventually calcium imaging how specific neuronal cell types respond to either mechanical cues or multi-modally integrate multiple cues. Overall, the paper represents an interesting investigation that is relevant for understanding how larvae select appropriate food sources. This paper adds to the increasing evidence that multi-modal neurons are common (at least in invertebrate models) and likely play an important role in foraging decisions. 

While the paper is overall convincing, I have some major and minor concerns detailed below. 

Major concerns:

- Overall, methods could be described in a touch more detail which would help the readers. I detail a few examples below where the lack of detail on what was quantified is confusing to the reader.

- decay is relative to ripeness etc., which the authors already acknowledge in the discussion. I suggest normalizing the compression modulus in panel A''' and/or adding a normalized panel as well as a sentence describing the limitation of estimating a 'start date' for the decay in the results section. In a similar vein, this sentence should also be rephrased: "This correlates to the hardness of pear and pineapple after 3-4 days of decomposition, highlighting that fruit hardness may provide sensory cues about the state of decomposition."

Ingestion assay: The ingestion assay should clarify if the % ingestion denotes the fraction of animals that showed blue dye (assessed visually?) or if this is a quantitative measure of the amount of ingested dye? This is unclear from the y-label and the results section.

Larval taste organ identification:

Fig. 2A - what are the gray points? Is this data from all head neurons or only DOG/VO?

My major concern about the paper is about the calcium imaging.

Live calcium imaging in Fig. 3:

I am very confused if the traces shown represent a mean or a sample and error bars are not shown? Without any error bars or quantification, we can not assess if the changes in intensity are meaningful, robust and significant!

The methods section for the calcium imaging does not describe how the data were analyzed, in neither the single cell or multi-cell imaging case. In addition, the authors should either make use of the ratiometric indicator to remove artifacts from changing the flow rate (which may move or compress the cells). Alternatively, please include a supplementary figure showing how the fluorescence in the red channel changes. This is particularly important for the mechanical stimulus, but should be presented for all calcium data, especially given the large spread in dF/F0 values in Figure 4. 

The term 'Live calcium imaging' for the data in Fig. 3 appears a bit misleading, as the preparation uses dissected heads. I suggest rephrasing this or at least acknowledging the difference between an intact animal and a dissection in the appropriate results section.

- In Fig. 4D, it seems the response of the RNAi of Gr63/Gr21 in C6 does have an effect in mechanosensation, namely the neuron seems to be inhibited much longer compared to wt (panel 4B). What is the interpretation of this difference (if it is significant)?

minor concerns:

Statistics: 

The data is overall very convincing and the below are points mainly to help the reader assess the work. 

In many figures it is hard to quickly see the sample sizes due to the color of the individual points. It would be helpful to have the N number denoted in the caption or figure itself, e.g., in Fig. 1B-B''', as well as for the calcium imaging in Fig. 4.

For Figure 1B, please describe what was tested e.g., based on the test it seems that group difference was assessed, rather than significant ingestion relative to 0. It would be useful to the reader to be explicit about this.

In Figure 2C', the authors state that larvae show a preference for softer substrates, however the chosen test (One-way ANOVA) tests for group-mean differences as indicated in the figure. To support this statement, the authors should test if the ablated data shows a significant preference compared to a preference index of 0. ('Larvae lose preference for the softer 1% agarose versus 209 harder 2.5% agarose, in addition they start to prefer excessively soft 0.1% agarose')

---

## [Decision Letter · Decision Letter 2]

24 Nov 2024

Dear Simon,

Thank you for your patience while we considered your revised manuscript "Food hardness preference reveals multisensory contributions of gustatory organs in behaviour and physiology" for publication as a Research Article at PLOS Biology. This revised version of your manuscript has been evaluated by the PLOS Biology editors, the Academic Editor and two of the original reviewers.

You'll see that both reviewers are happy with your revisions. Based on the reviews and on our Academic Editor's assessment of your revision, we are likely to accept this manuscript for publication, provided you satisfactorily address the following data and other policy-related requests.

IMPORTANT - please attend to the following:

a) Please change your Title to "Food hardness preference reveals multisensory contributions of fly larval gustatory organs to behavior and physiology"

b) Please address my Data Policy requests below; specifically, we need you to supply the numerical values underlying Figs 1A'BB'B''CC'C'', 2ACC', 3ABB'DE, 4ABCDD', S1B'C, S2ABCDEF, S3AB, S4ABC, either as a supplementary data file or as a permanent DOI’d deposition.

c) Please cite the location of the data clearly in all relevant main and supplementary Figure legends, e.g. “The data underlying this Figure can be found in S1 Data” or “The data underlying this Figure can be found in https://zenodo.org/records/XXXXXXXX

d) Please make any custom code available, either as a supplementary file or as part of your data deposition.

We expect to receive your revised manuscript within two weeks. 

*Published Peer Review History*

*Press*

Sincerely,

Roli

Roland Roberts, PhD

Senior Editor

rroberts@plos.org

PLOS Biology

DATA POLICY:

Regardless of the method selected, please ensure that you provide the individual numerical values that underlie the summary data displayed in the following figure panels as they are essential for readers to assess your analysis and to reproduce it: Figs 1A'BB'B''CC'C'', 2ACC', 3ABB'DE, 4ABCDD', S1B'C, S2ABCDEF, S3AB, S4ABC. NOTE: the numerical data provided should include all replicates AND the way in which the plotted mean and errors were derived (it should not present only the mean/average values).

CODE POLICY

DATA NOT SHOWN?

REVIEWERS' COMMENTS:

Reviewer #3:

The authors have adequately addressed my concerns and I therefore recommend that the manuscript be published the manuscript as it stands in PLOS Biology.

Reviewer #4:

The authors of the manuscript now entitled 'Food hardness preference reveals multisensory contributions of gustatory organs in behaviour and physiology' have revised the manuscript in response to mine and the other reviewers comments. I find that the revised manuscript addressed my prior concerns fully, and thus should be accepted. 

Congratulations to the authors on a nice paper!

---

## [Editor Report · Decision Letter 3]

5 Dec 2024

Dear Simon,

Thank you for the submission of your revised Research Article "Food hardness preference reveals multisensory contributions of fly larval gustatory organs in behaviour and physiology" for publication in PLOS Biology. On behalf of my colleagues and the Academic Editor, Matthieu Louis, I'm pleased to say that we can in principle accept your manuscript for publication, provided you address any remaining formatting and reporting issues. These will be detailed in an email you should receive within 2-3 business days from our colleagues in the journal operations team; no action is required from you until then. Please note that we will not be able to formally accept your manuscript and schedule it for publication until you have completed any requested changes.

IMPORTANT: Many thanks for providing the folder called “Charts” in your Zenodo deposition. I see that this contains a single spreadsheet called “RawData” – this has the underlying values of Figs 1-3, but I note that the sheets for Figs 4 and S1-S4 are empty, which is presumably an error - please could you complete this file before publication?

Sincerely, 

Roli

Senior Editor

PLOS Biology

rroberts@plos.org